# Lifestyle and Chronic Pain in the Pelvis: State of the Art and Future Directions

**DOI:** 10.3390/jcm10225397

**Published:** 2021-11-19

**Authors:** Annelie Gutke, Karin Sundfeldt, Liesbet De Baets

**Affiliations:** 1Department of Health and Rehabilitation, Institute of Neuroscience and Physiology, Sahlgrenska Academy, University of Gothenburg, 40350 Gothenburg, Sweden; 2Department of Obstetrics and Gynecology, Institute of Clinical Sciences, Sahlgrenska Academy, University of Gothenburg, 40350 Gothenburg, Sweden; karin.sundfeldt@obgyn.gu.se; 3Department of Gynecology, Sahlgrenska University Hospital, 41346 Gothenburg, Sweden; 4Pain in Motion Research Group (PAIN), Department of Physiotherapy, Human Physiology and Anatomy, Faculty of Physical Education & Physiotherapy, Vrije Universiteit Brussel, 1050 Brussel, Belgium; liesbet.de.baets@vub.be

**Keywords:** chronic pelvic pain, endometriosis, pelvic girdle pain, lifestyle factors, pain management, physical activity/exercise, (di)stress, sleep, diet, smoking

## Abstract

During their lifespan, many women are exposed to pain in the pelvis in relation to menstruation and pregnancy. Such pelvic pain is often considered normal and inherently linked to being a woman, which in turn leads to insufficiently offered treatment for treatable aspects related to their pain experience. Nonetheless, severe dysmenorrhea (pain during menstruation) as seen in endometriosis and pregnancy-related pelvic girdle pain, have a high impact on daily activities, school attendance and work ability. In the context of any type of chronic pain, accumulating evidence shows that an unhealthy lifestyle is associated with pain development and pain severity. Furthermore, unhealthy lifestyle habits are a suggested perpetuating factor of chronic pain. This is of specific relevance during lifespan, since a low physical activity level, poor sleep, or periods of (di)stress are all common in challenging periods of women’s lives (e.g., during menstruation, during pregnancy, in the postpartum period). This state-of-the-art paper aims to review the role of lifestyle factors on pain in the pelvis, and the added value of a lifestyle intervention on pain in women with pelvic pain. Based on the current evidence, the benefits of physical activity and exercise for women with pain in the pelvis are supported to some extent. The available evidence on lifestyle factors such as sleep, (di)stress, diet, and tobacco/alcohol use is, however, inconclusive. Very few studies are available, and the studies which are available are of general low quality. Since the role of lifestyle on the development and maintenance of pain in the pelvis, and the value of lifestyle interventions for women with pain in the pelvis are currently poorly studied, a research agenda is presented. There are a number of rationales to study the effect of promoting a healthy lifestyle (early) in a woman’s life with regard to the prevention and management of pain in the pelvis. Indeed, lifestyle interventions might have, amongst others, anti-inflammatory, stress-reducing and/or sleep-improving effects, which might positively affect the experience of pain. Research to disentangle the relationship between lifestyle factors, such as physical activity level, sleep, diet, smoking, and psychological distress, and the experience of pain in the pelvis is, therefore, needed. Studies which address the development of management strategies for adapting lifestyles that are specifically tailored to women with pain in the pelvis, and as such take hormonal status, life events and context, into account, are required. Towards clinicians, we suggest making use of the window of opportunity to prevent a potential transition from localized or periodic pain in the pelvis (e.g., dysmenorrhea or pain during pregnancy and after delivery) towards persistent chronic pain, by promoting a healthy lifestyle and applying appropriate pain management.

## 1. Introduction

During their lifespans, women are at a high risk for experiencing pain complaints in the pelvic region due to gynecologic and obstetric reasons. Indeed, hormonal changes in the menstruation cycle are typically associated with pain in the pelvis [1]. Most women experience pain for one or more days of the menstruation period, for which pain killers or hormonal contraceptive pills are the first-choice treatment. However, this pharmacological treatment is not sufficient for pain reduction in an important subgroup of women [2], resulting in recurrent or persistent pelvic pain, such in women who experience endometriosis-related pelvic pain [3]. Severe dysmenorrhea (pain during the menstrual period) is a cardinal symptom of endometriosis and is known to have a high social impact as it is often associated with absence from school or work [1]. Apart from endometriosis-related pelvic pain, pain in the pelvis also occurs in relation to pregnancy and childbirth [4]. Such pregnancy-related pelvic girdle pain (PGP) is suggested to relate to both hormonal, musculoskeletal and biomechanical changes [4,5]. The pain intensity is often severe enough to hinder pregnant women or women in the postpartum period to participate in activities of daily living, including work [4,6,7,8]. Since pharmacological treatment is not the first choice for pain management for many pregnant or breast-feeding women, other conservative approaches, such as physical therapy, are generally applied [4,9]. However, this care is not considered successful in appropriately alleviating pain in a subgroup of women with PGP, or is not consistently offered to women with PGP [10].

Indeed, women experiencing pain in the pelvis (endometriosis-related pelvic pain and pregnancy-related PGP) are not consistently guided towards appropriate treatment for their pain complaint. It is even more disturbing that these women are often told that their experienced pain should be considered ‘normal’, and inextricably linked to being a woman [3,11]. Apart of being unethical, such a message also contributes to the fact that many women seek too little help for a pain complaint that may be treatable [10]. This might result in the fact that an initial pain experience, e.g., experienced during menstruation or pregnancy or immediately following childbirth, evolves towards persistent pain in the pelvis.

To improve this current practice, knowledge on adequate treatment approaches for reducing pain in women with pain in the pelvis is required. It is furthermore believed that adequate strategies to prevent an increase in pain during periods of life in which a hormone or pregnancy-related pain can be foreseen, are of even greater importance. In this context, the effect of lifestyle changes on pain has received more attention in recent years. There are many possible causes for pain in the pelvis, and in this review we focus on two common disorders related to specific painful events during lifespan of women to point out the possibility to prevent development of chronic pain. However, to understand why a healthy lifestyle might be effective in reducing pain in women with endometriosis-related pelvic pain and pregnancy-related PGP, more information about the pathophysiology of these conditions is first provided. 

### 1.1. Endometriosis-Related Chronic Pelvic Pain

Endometriosis is, with a reported prevalence around 10%, a very common condition among women of childbearing age, and one of the most common structural causes of chronic pelvic pain [3,12]. It is classically defined as an estrogen-dependent, chronic inflammatory condition in which the endometrium of the uterus grows outside the uterus by implantation of endometrial cells and creates an inflammatory response of the surrounding tissue. Both superficial lesions on the peritoneum, ovarian endometriosis and a more aggressive deep infiltrating type that may obstruct the intestines or bladder, have been reported [13]. In this context, increasing evidence suggests that endometriosis should be considered a systemic disease, not only restricted to the pelvis [3]. Indeed, endometriosis is reported to co-exist with other conditions such as irritable bowel syndrome, mental health disorders, central sensitization and pain conditions (fibromyalgia, migraine), and immunological conditions (e.g., rheumatoid arthritis, multiple sclerosis) [13,14]. Given these multiple facets of endometriosis, its diagnosis is currently very challenging [13], resulting commonly in a delay in diagnosis of multiple years [3,13,15,16].

Variable clinical symptoms are related to endometriosis. Typical pain symptoms in women with endometriosis are (a)cyclic pelvic pain, dysmenorrhea and pain at ovulation [3,13,15]. This may be combined with radiating pain to the lower back, groin and thighs, as well as deep intercourse pain [3]. Bladder problems, difficulties in emptying the bowel and infertility can be attributed to endometriosis as well [17]. Given these debilitating symptoms, endometriosis might affect a woman’s physical, mental and social well-being, her quality of life and her work ability [18]. These, in turn, are related to higher levels of psychological distress. Furthermore, poorer sleep quality, endometriosis-related fatigue and physical deconditioning are reported in women suffering from endometriosis [19].

Current first-line treatment for endometriosis is conservative pharmacological care, and surgical procedures when pharmacological options are ineffective or with deep infiltrating endometriosis. However, for a large group of women, endometriosis recurs after surgical treatment or are ineffective, as evident from the large rate of (50%) [3]. In this context, the link between the grade of laparoscopic detected lesions and level of symptoms is also inconsistent [3]. This emphasizes the value of looking beyond tissue-related aspects and to look for treatment options addressing the mechanisms underlying symptom development. Therefore, it seems valuable to take a closer look into the lifestyle of women suffering symptomatic endometriosis. As endometriosis is considered an inflammatory and estrogen-dependent disease, targeting lifestyle factors that influence these factors can open the path for multiple conservative treatment options for women with endometriosis-related pelvic pain (e.g., dietary intervention, physical activity, sleep management). Thereby, the anti-inflammatory effect of a healthy lifestyle might positively influence pain perception given the association between inflammatory mediators and peripheral as well as central sensitization [20].

### 1.2. Pregnancy-Related Pelvic Girdle Pain

Pregnancy-related PGP is reported by 50% of pregnant women globally [4,5,10]. PGP is classified by its pain location, i.e., between the posterior iliac crest and the gluteal fold, most commonly around the sacroiliac joints and/or pubic bone, sometimes with radiating pain in the thighs, with onset close to, or within three weeks of delivery [5]. For the classification of PGP, it is recommended that the pain is reproduced during clinical pain provocations tests and that it is associated with, and time-dependent on, weight bearing activities. Typical symptoms of PGP, both during and after pregnancy are a decrease in endurance in standing, walking, and sitting (often within 30 min of activity) [4,21] which lead to limitations in daily functioning [6,7,8], and at work [4,22]. Since PGP is mostly related to pregnancy, it is expected to disappear when pregnancy-related changes disappear after birth [6], which is true for the majority of women [23]. However up to 11 years after pregnancy, 10% of women report persistent and per definition chronic PGP [8]. The etiology of PGP is multifactorial. One suggested cause of PGP is inefficient neuromuscular control [4,5] related to hormonal changes during pregnancy. However, a systematic review of evidence on whether this results in pelvic instability causing PGP is low [24]. In women with PGP, higher prevalence rates of prenatal anxiety and depressive symptoms are reported in comparison to pregnant women without pain [25,26]. This co-occurrence of pain with anxiety and depression continues in women with chronic PGP, who also seem to have less general self-efficacy than women who recovered from PGP after pregnancy [8]. Furthermore, from a recent cohort study [27], lack of physical activity was added as a predisposing factor for pain in the lumbo-pelvic area in pregnancy together with the hitherto reported previous history of pain in the lumbo-pelvic area, low job satisfaction, and increased weight during pregnancy [4]. 

In general, the evidence is low for the effectiveness of interventions for PGP [4,9]. As PGP treatment, acupuncture shows the most coherent findings in the literature [4,9,28]. Other suggested treatment strategies are the application of a pelvic belt and exercises [4,9]. In recent years, a cognitive behavioral approach and self-management strategies have been proposed for women with PGP [29]. In the above-described context, it seems valuable to take a closer look into the lifestyle factors of women suffering from PGP.

### 1.3. Lifestyle Factors in Chronic Pain in the Pelvis

Within the chronic pain field, there is cumulating evidence that unhealthy lifestyle factors such as physical inactivity, increased psychological distress, poor sleep, unhealthy diet, and smoking are associated with chronic pain severity and sustainment [30,31,32]. A proposed mechanism underlying the association between lifestyle and pain in this regard is related to inflammatory mediators. For example, it is known that in chronic pain patients, disturbed sleep modulates the endogenous inhibitory pain control system, produces changes in the hypothalamic-pituitary-adrenal axis and induces aberrant inflammatory reactions [33]. Specifically in relation to pregnancy, an increase in inflammatory biomarkers in sleep-deprived pregnant women in comparison to non-pregnant has been reported [34]. Even after pregnancy, an inflammatory response with increased cytokine levels is associated with short sleep duration [35]. 

Within the field of pain in the pelvis, the scientific literature points towards the role of lifestyle on pain. Being overweight/obese or experiencing emotional distress during pregnancy have been associated with less recovery at 3–6 months after pregnancy when the natural course of PGP is over [36]. In women with chronic PGP, level of physical activity, exercises, sleep quality and distress are associated with pain perception [8]. Regarding lifestyle factors and endometriosis, poor sleep quality is reported to be associated with pelvic pain [37]. 

An increased understanding of the role of exercise, insomnia, diet, and other lifestyle factors on the perception of pain in women with pain in the pelvis is assumed to provide treatment opportunities to optimize current care. Of even greater importance, such understanding on the role of lifestyle factors in the development and maintenance of pain in the pelvis, can provide invaluable knowledge on how we might develop prevention strategies for severe pain in the pelvis. This is of utmost importance, since it can be theorized that painful experiences early in a woman’s life that are not handled according to the best of knowledge, might be a reason why higher rates of chronic pain in women as compared to men are reported [38,39]. Therefore, in the early stages, when pain in the pelvis can be expected, there is a window of opportunity to take the right preventive actions, with good pain management strategies, and possibly by advocating a healthy lifestyle.

This review aims to give an overview of the best evidence on the role of lifestyle factors in the development or maintenance of pain in the pelvis in women related to common painful events during their lifespan. Since very few studies are published in the field, a systematic review was not considered feasible. A best-evidence review was considered the appropriate format to explore the field and to present a research agenda. A best-evidence synthesis of the effect of lifestyle interventions on women with pain in the pelvis is provided. The best evidence knowledge is reviewed in a way such that clinicians can integrate the evidence into their daily clinical routine. In addition, the state-of-the-art overview also serves clinical researchers in building upon the best evidence for designing future trials, implementation studies, and to develop new innovative studies.

## 2. State-of-the-Art

For this best-evidence review, the following lifestyle factors were defined a priori: physical (in)activity, exercise, sleep, psychological distress, food intake, tobacco use, smoking and alcohol consumption. With regard to the interventions for these lifestyle factors, only active interventions such as exercise, psychologically informed approaches, cognitive behavioral therapy, dietary interventions (focusing on altering food uptake) and multimodal approaches, were considered. Studies on surgical procedures, pharmaceutical treatment in isolation (including supplements e.g., vitamin supplements), Chinese medicine and passive treatments such as acupuncture, in isolation, were not eligible. 

A nonsystematic search of scientific studies was performed in MEDLINE (PubMed), and web-of-science from their inception to August 2021, using the following search terms: ((endometriosis OR pelvic girdle pain OR lumbopelvic pain) AND (physical activity OR exercise* OR insomnia OR sleep OR stress OR diet OR nutrition OR smoking OR tobacco OR alcohol). The searches were conducted by two researchers (AG and LDB) independently. To minimize selection bias and to ensure high quality evidence was selected, systematic reviews and meta-analyses in accordance with PRISMA guidelines were preferred. If these were not available, narrative, and critical reviews were selected. Recent high-quality prospective studies and randomized clinical trials (RCTs) not already included in systematic reviews were also included, as well as information from large population-based cohorts.

Physical activity and exercise were defined as any bodily movement generated by skeletal muscles resulting in energy expenditure above resting levels [40]. The use of the terms ‘physical activity’ and ‘exercise’ in the included studies was not consistently in accordance with published definitions [40], and the terms were sometimes used interchangeably. Therefore, this review was not able to differentiate between interventions for either of them. 

To retrieve all relevant articles within the area of PGP, a search was done for both PGP and lumbopelvic pain (LPP), since the latter term is often used when the studies do not distinguish between PGP and combined pain from the lumbar and pelvic areas [41]. Some authors have used the term low back pain (LBP) for pain in the lumbar area and evaluated the subgroup with LBP separately e.g., Weis et al., [42], which is also presented separately in our review.

## 3. Endometriosis-Related Chronic Pelvic Pain and Lifestyle Factors

Seven systematic reviews were found that were prepared and outlined in accordance with the PRISMA guidelines. Of these reviews, four provided a narrative synthesis of their data due to study heterogeneity (design, research questions, outcomes, interventions, etc.). Three studies performed a meta-analysis. Characteristics of the included systematic reviews and meta-analyses, together with their main results and level of evidence (if applicable) are outlined in Table 1. 

In the text that follows, the results of the systematic reviews and meta-analyses are described, together with the results of narrative/critical reviews and recent original studies.

### 3.1. Physical Activity and Exercise

Two systematic reviews reported on endometriosis and physical activity/exercise. Hansen and colleagues (2021) studied the recent evidence on the impact of exercise on pain perception in women with endometriosis [43]. No general positive effect of exercise on pain could be concluded. However, the included studies were generally of high risk of bias. Ricci et al., (2016) studied the role of physical activity on the risk for endometriosis [44]. In their meta-analysis, women with endometriosis performing recent physical activity, and women performing physical activity in the past, were included. The pooled estimate of adjusted odds ratios for current exercise indicated a significantly protective effect of exercise, but the overall estimates did not reach levels of significance. Furthermore, the review did not specify the influence of physical activity on pain symptoms. The aforementioned results are in line with the results of the earlier published narrative review of Bonocher et al., (2014) that assessed the relationship between physical exercise and the prevalence and/or improvement of symptoms associated with endometriosis [50]. 

The data available are inconclusive regarding the benefits of physical exercise on the risk of endometriosis, and no firm data exist on the added value of physical activity on pain in women with endometriosis. 

### 3.2. Psychological Distress

One systematic review was found that assessed the effectiveness of psychological and mind-body interventions to improve pain, psychological distress, sleep and fatigue in women with endometriosis [48]. The studies assessed the value of yoga, mindfulness, relaxation training, cognitive behavioral therapy combined with physical therapy, Chinese medicine combined with psychotherapy, and biofeedback. No firm conclusions could be drawn given the high variety of interventions and designs. Most studies were considered pilot studies. However, the results of the studies suggested that psychological and mind-body interventions are promising avenues to decrease pain, anxiety, depression, distress, and fatigue in women with endometriosis.

### 3.3. Sleep

No systematic nor narrative reviews on the role of insomnia on pain in women with endometriosis were found, nor on the effectiveness of sleep interventions on pain symptoms in women with endometriosis. No recent original prospective cohort studies or intervention studies regarding this topic were found. 

Arion et al., (2020) performed a quantitative analysis of sleep quality in women with surgically confirmed endometriosis to assess which variables were associated with poorer sleep [51]. Based on regression analyses, the following factors were independently associated with poorer sleep: functional quality of life, more depressive symptoms and painful bladder syndrome. In a former cross-sectional study on the sleep quality of women with endometriosis and the relation between sleep quality and pressure pain thresholds, sleep quality was significantly poorer in women with endometriosis compared to women without endometriosis [52]. Furthermore, the pressure pain threshold in the greater trochanter and abdomen was significantly lower in women with endometriosis when compared to women without endometriosis, which is indicative of an increased central sensitivity; however, there was no difference in pain intensity between women with and without endometriosis.

### 3.4. Diet

Huijs and Nap (2020) performed a literature search to gain insights into the role of nutrients on the symptoms of women with surgically or magnetic resonance imaging/ultrasound confirmed endometriosis [45]. Using the GRADE criteria, the quality of the evidence in this review turned out to be low to very low. It was suggested that the intake of additional fatty acids, antioxidants and a combination of vitamins and minerals could have a positive effect on endometriosis-associated symptoms. 

Nirgianakis et al., (2021) performed a systematic review on the effectiveness of dietary interventions in the treatment of endometriosis [46]. Changes in endometriosis-associated symptoms measured with pain scales or patient-reported quality of life outcomes were the outcomes of interest in this systematic review. Different dietary interventions were assessed, including: supplementation of vitamin D; supplementation of vitamins A, C, and E; supplementation of omega-3/6, quercetin, vitamin B3, 5-methyltetrahydrofolate calcium salt, turmeric, and parthenium; Mediterranean diet; low-FODMAP diet; low nickel diet; gluten-free diet, and individual diet changes. Most studies identified a positive effect of the dietary intervention on endometriosis symptoms. However, all studies were of moderate and/or high-risk risk of bias limiting the validity of the results. Furthermore, it was not possible based on the available evidence to identify certain subcategories of patients, which would be more likely to benefit from a dietary intervention. In addition, it was not possible, based on the current literature, to identify specific dietary interventions that would ameliorate certain endometriosis-associated symptoms. Therefore, it was concluded that more, and especially higher quality original studies, are needed to draw conclusions on the effectiveness of dietary intervention on pain in women with endometriosis.

Recently, Qi et al., (2021) performed a systematic review and dose-response meta-analysis to investigate the association between dairy products and the risk of endometriosis, and to evaluate the amount of dairy intake affecting the risk of endometriosis [47], though the effect on pain perception was not taken into consideration. Based on the meta-analysis results, the authors concluded that the intake of dairy products was associated with a reduction in endometriosis when the average daily intake was three servings or more. When analyzed according to the specific type of dairy product, it was suggested that females with a higher high-fat dairy and cheese intake were at lower risk of endometriosis. Regarding butter intake, it was suggested that high intake was related to an increased risk of endometriosis. All studies included in this meta-analysis were of high quality, as based on the Newcastle-Ottawa Scale [53].

Additionally, Helbig et al., (2021) performed a literature search for articles from 2000 onwards to answer the question whether diet influences the risk for and progression of endometriosis or whether it influences the postoperative condition [54]. This review did not take the effect of diet on pain symptoms into account. Based on the evidence, it was suggested that fish oil capsules in combination with vitamin B12 were associated with a positive effect on endometriosis symptoms (particularly of dysmenorrhea). It was reported that alcohol and increased consumption of red meat and trans-fats were associated with a negative effect on endometriosis. The results of the studies listed with regard to fruit and vegetables, dairy products, unsaturated fats, fibre, soy products and coffee were not clear.

In conclusion, no high-qualitative prospective data on the role of diet (which food products and in which amounts) in the development and maintenance of pain in women with endometriosis are available. Furthermore, no firm data exist on the added value of dietary interventions on pain in women with endometriosis, given the low study quality of currently existing trials.

### 3.5. Tobacco/Alcohol Use

One systematic review and meta-analysis studied the relation between tobacco smoking and endometriosis risk. No evidence for an association between tobacco smoking and risk of endometriosis was found [49]. When subgroups were considered, i.e., never smokers vs. former smokers, current smokers, moderate smokers or heavy smokers, no statistically significant associations were reported.

## 4. Pelvic Girdle Pain and Lifestyle Factors

Seven systematic reviews were found that evaluated lifestyle intervention for PGP prepared and outlined in accordance with the PRISMA guidelines. Of these reviews, five studies performed a meta-analysis. Characteristics of the included systematic reviews and meta-analyses, together with their main results and level of evidence are outlined in Table 2. 

In the text that follows, the results of the systematic reviews and meta-analyses are described, together with the results of narrative/critical reviews and recent original studies.

### 4.1. Physical Activity and Exercise

Seven systematic reviews, performed according to PRISMA, reported on physical activity/exercise as management of PGP. Weis et al., (2020) conducted two systematic reviews on systematic reviews and RCTs to assess the effectiveness of chiropractic care options including exercises commonly used for pregnancy-related PGP, LPP and LBP during pregnancy [55] and postpartum [42]. For PGP in pregnancy, there was inconclusive evidence that an exercise program was more effective to decrease pain and disability compared with standard treatment [55]. For LPP, it was found that exercise had unclear outcomes on improvements in function (moderate strength evidence), and that exercise reduced the prevalence of LPP. This latter was reported in most included systematic reviews. For LBP, studies had inconclusive strength evidence with favorable outcomes on decreased pain and disability. From a systematic review focusing on the postpartum period [42], there was moderate evidence with no clear outcomes to suggest exercise as treatment for or prevention of PGP, since only one of three included systematic reviews stated the additional effect of exercises at reducing pain and disability. No firm conclusion could be drawn for LPP, but the authors reported some evidence which indicated that exercises could relieve LPP. There were no results for LBP postpartum. Davenport et al., (2019) performed a systematic review to investigate the relationship between the performance of prenatal exercises and PGP, LPP as well as LBP [56]. Based on very low-to-moderate quality evidence, prenatal exercise compared to no exercise during pregnancy did decrease pain severity in pregnancy and at postpartum but did not reduce the odds of suffering from PGP, LPP and LBP either in pregnancy or at postpartum. 

Almousa et al., (2018) evaluated the effectiveness of stabilizing exercises in PGP during pregnancy and postpartum in a systematic review [57]. They concluded that there was limited evidence that stabilizing exercises decreased pain and improved quality of life during pregnancy and postpartum. Shiri et al., (2018) did a meta-analysis of RCTs to study the value of exercise in the prevention of PGP and LBP [58]. They concluded that exercise reduced the risk of LBP in pregnancy by 9% but that exercise had no effect on PGP or LPP. Additionally, exercise prevented new episodes of sick leave due to LPP. Liddle and Pennick (2015) performed a systematic review according to Cochrane Collaboration’s tool to update the evidence on the effects of any intervention including exercise to prevent or treat PGP, LPP or LBP during pregnancy [4]. Low-quality evidence showed no significant difference in the prevalence of PGP or LBP from exercise. Low-quality evidence showed that any land-based exercise significantly reduced pain and disability from LBP. Moderate-quality evidence showed reduced the prevalence of LPP from 8–12 weeks of exercises. 

Tseng et al., (2015) aimed to synthesize evidence from RCTs on the effectiveness of exercise on LPP in postpartum women [59]. Based on four RCTs they concluded that there was some evidence to indicate the effectiveness of exercise for relieving LPP but more trials were need to ascertain the most effective postpartum exercise programs. 

Even though not proven to reduce the risk of developing PGP [56,58], a recent systematic review on physical activity and exercise in relation to pregnancy reported decreased severity of PGP and LPP [56]. Davenport et al. reported their intervention results separated as ‘exercise only’ or ‘exercise plus co-intervention’ [56], which could explain the somewhat different result to another recent systematic review that reported inconclusive, although favorable, evidence for exercises in pregnancy [55] as well as at postpartum [42]. These latest results confirm the most recent Cochrane review on exercise in pregnancy for LPP and LBP, but not the previous result of no effect for PGP [4]. The newer result, that exercise can reduce the severity of PGP, may be explained by more studies to build evidence on. Likewise, the number of studies to build the evidence on is probably the explanation of some other previously limited [57], inconclusive evidence [59] reported. The recent results confirm an earlier narrative review on LBP and PGP by Stuge (2015), where it was concluded that there is evidence of moderate quality that exercise reduced pain intensity but not prevalence [60].

Various types of exercises including general physical activity, low impact aerobic exercise such as walking, stabilizing exercises, resistance training, and other forms of exercises such as Yoga, or the combination of different exercises, have been evaluated for their effectiveness in PGP, LPP and LBP in the included systematic reviews. Systematic reviews usually include all exercise types together. Only one identified systematic review differentiated exercise only from exercise with co-interventions [56], but no significant difference between pooled estimates could be seen (pregnancy *p* = 0.24; postpartum *p* = 0.70). At this time, there is insufficient evidence to determine whether one type of exercise is superior to another or whether exercise should be combined with other interventions and, in that case, which co-intervention(s). One identified systematic review that focused on group training for LPP reported no effect as treatment of LPP among pregnant women but reported group training to be effective after pregnancy [61]. 

From four population-based cohort studies in Brazil, it was reported that 41.9% of 3827 pregnant women reported LBP of any type, and 10% of women during pregnancy reached recommended levels of physical activity [62]. The authors concluded that meeting the recommended levels of physical activity during pregnancy was associated with less activity limitation related to LBP during pregnancy. However, physical activity levels, either before (β coefficient: 0.07; 95% CI, −0.25 to 0.38) or during pregnancy (β coefficient: −0.07; 95% CI, −0.46 to 0.33), were not associated with pain intensity, care seeking, and postpartum LBP [62]. After pregnancy, physical activity level was continuously suboptimal for many women [63]. Recently, it was reported that sedentary behavior after birth was associated with persistent LPP in primiparas, but not multiparas [64]. The authors interpreted this to be a result of multiparas needing to be more active when raising an older child.

### 4.2. Psychological Distress

No systematic nor narrative reviews on the effectiveness of (di)stress interventions on pain symptoms in women with PGP were found. 

In a recent systematic review on prognostic factors, experiencing emotional distress during pregnancy was associated with less recovery and severe pelvic girdle syndrome at 6 months after pregnancy, when the natural course of PGP is over (*n* = 40,029; Adjusted Odds Ratio (AOR) 1.3, 95% CI (1.1–1.5) [36]. The findings need to be taken with caution since quality of evidence according to GRADE was low to very low. Distress during pregnancy has also been associated with chronic PGP after delivery [65]. The conclusion is that there are indications that distress can affect PGP and its course and that this needs to be studied in more detail both within a prevention as well as a curative context.

### 4.3. Sleep

There was no identified systematic review regarding the effect of sleep management on PGP. Among pregnant women with pain in the lumbopelvic area, a high-pain group reported worse emotional health and poorer sleep quality than controls without pain [66]. Sleep disturbance has also been associated with persistent PGP and LPP at 4 months after pregnancy, even after adjustment for possible confounding variables such as BMI, parity, age, and history of LPP [67]. Sleep impairment related to quantity and adequacy of sleep has been associated with women with moderate disability from persistent PGP after pregnancy [68]. Different to these results, in a recent study the moderate or severe sleeping complaints associated with PGP disappeared after adjustment for depression [69]. This might be explained by the comorbidity of PGP and depression [26]. Many women experience disturbed sleep during pregnancy due to continuous hormonal-related changes of the body, need of nocturia and movement from the fetus. It is also known that around 50% of women get disturbed sleep during the first years of parenthood due to nighttime feeding and nocturnal awakening among infants [70,71]. 

To conclude, there is no evidence of sleep interventions for PGP. However, concerning the prevalence of sleep disturbance in relation to pregnancy and early childhood of women, there are indications of the importance of sleep as a lifestyle factor in women with chronic PGP. 

### 4.4. Diet

There was no identified systematic review on the effectiveness of dietary interventions in the treatment of PGP. However, if an unhealthy diet leads to a high body mass index (BMI) there are some associations to consider. A systematic review reported an association between a BMI of 25 or more and having persistent PGP 12 weeks after pregnancy (*n* = 179; AOR 2.1, 95% CI 1.0–4.5) [36]. From the same study, it was reported that obese women (BMI 30 or more) had higher odds to have persistent pelvic girdle syndrome (*n* = 27,025; AOR 1.8, 95% CI 1.5–2.0) and severe pelvic girdle syndrome at 6 months (*n* = 27,025; AOR 1.6, 95% CI 1.1–2.4). Pre-pregnancy BMI > 25 was shown as a risk factor for chronic PGP in another recent systematic review [72]. Since interaction between factors could not be done in a multivariate analysis, the results should be taken with care [72]. 

Lifestyle intervention of physical activity in general in relation to pregnancy and postpartum are often focused on BMI as an outcome [73,74]. Effectiveness of a combination of diet and physical activity consistently showed a reduction in mean gestational weight gain during pregnancy and postnatal weight retention [75]. There were no reports on musculoskeletal pain of the studied groups, but from a coherent literature on PGP it is can be assumed that about 50% of the women had PGP in pregnancy and around 25% had persistent PGP after birth [4]. A major part of women used walking as physical activity intervention in the studies that reported types of physical activity [76,77]. Interesting to consider is whether professional individualized exercise advice would have further improved the outcomes.

Although no systematic review on PGP and diet was identified, there seems to be promising results of intervention for a combination of diet and physical activity in relation to pregnancy that needs further exploration of possible effects on chronic PGP.

### 4.5. Tobacco/Alcohol Use

There was no identified systematic review regarding the effect of tobacco or alcohol interventions on PGP.

In a systematic review on prognostic factors, Wuytack et al. reported an association with occasional smoking (*n* = 38,865; AOR 1.3, 95% CI 1.0–1.6), but not daily smoking, in women with pelvic girdle syndrome at 6 months postpartum [36]. 

## 5. Future Directions for Clinical Practice

In line with the World Health Organization’s (WHO), a stronger focus towards a healthy lifestyle is recommended, i.e., being physically active for better health and for preventing noncommunicable diseases such as diabetes, cancer, and cardiovascular disease [78]. In the action plan for the prevention of long-term disability from LBP [79] and chronic pain in general across the lifespan [80], a positive health concept as an overarching strategic approach is emphasized. In this, a positive health concept entails, among others, learning to cope with a chronic health problem through self-management strategies. A healthier lifestyle promoting physical activity and staying active despite pain, maintaining a healthy weight, and promoting mental health, are in this context primary prevention strategies for chronic disability [79,80]. When specifically looking into the results of this state-of-the-art paper on lifestyle factors and their role in women with pain in the pelvis, only a few recommendations can be made based on the available literature. Regarding physical activity and exercise, encouraging women to be physically active and to exercise is supported to some extent. Nonetheless, because physical activity and exercises of different intensity, frequency, duration, and type were used in the available evidence, no specific recommendations regarding this point can be made. The available evidence on lifestyle factors such as sleep, (di)stress, diet, and tobacco/alcohol use for women with pain in the pelvis is inconclusive, since very few studies are available, and the studies which are available are of general low quality. Furthermore, most studies did not focus on the role of targeting lifestyle factors with the aim of improving pain in the pelvis. Therefore, no specific recommendations on the application of management strategies for these lifestyle factors can be provided. 

We can, however, approach pain management in women with pain in the pelvis from a modern pain management point of view, which includes the management of lifestyle factors. Indeed, modern pain management implies change in the focus from pain reduction to pain management, i.e., managing thoughts and feelings related to pain (i.e., catastrophic worry), thereby influencing knowledge about pain [81,82,83] and providing opportunities for behavioral change towards a more active and healthier lifestyle. It also implies accurate self-management, which supports autonomy, and includes educational and supportive interventions to increase the skills and confidence for persons in pain to manage their health problems [84]. 

For women who are prone to pain experiences early in life, as related to menstruation [85] and pregnancy [4], learning healthy pain management is a priority. This includes the assessment and management of the individual woman with pain in the pelvis, taking into account her history, her present context and framing her messages into bio-psycho-social and bio-inflammatory-psychological perspectives [29,86,87,88,89,90]. In this context, it is important for women who experience pain (cyclic pain from menstruation, local pregnancy-related pain, persistent pain at postpartum) to learn to approach activity despite pain. Indeed, physical activity has been described as a way to achieve exercise-induced analgesia [85,91] and as a strategy to promote self-efficacy (because of the experience of self-control [92], which is highly important given the relation between low self-efficacy and the development of disability [84]. In the context of pain in the pelvis, it has been shown that women with chronic PGP have less general self-efficacy than women who recovered from PGP after pregnancy [8]. From chronic pain science, it is known that physical activity additionally might have sleep improving [93], stress-reducing [94] and general anti-inflammatory effects [95], which are all relevant for optimal pain management [96]. Importantly, the intensity, volume, duration and type of activity that is most appropriate to reach these effects in women with pain in the pelvis are not firmly studied [97]. 

Nonetheless, it is known that pain is considered an important barrier against physical activity in women with pain in the pelvis [98]. For example, it is reported that women with PGP during pregnancy are less likely to exercise regularly [99]. Walking is the most chosen physical activity during pregnancy [77], but it is also the most painful physical activity when suffering from PGP [9]. Thus, decreased physical activity in pregnancy could be partly explained as a consequence of PGP. Importantly, women need to be guided into alternative physical activities and exercises when walking is painful. Therefore, in order to motivate women with pain in the pelvis to uptake physical activity despite tolerable levels of pain, and to reach a behavioral change, the use of a person-centred approach is essential [89]. A person-centred approach may increase the alliance between the care provider and the woman with pain in the pelvis, and subsequently lead to better adherence, improvements in general health and satisfaction [100]. It is important to approach the worries and concerns related to physical activity and exercise, e.g., how to interpret pain-increasing physical activity, which is different for a nociceptive versus nociplastic dominant pain mechanism. Pain neuroscience education is a known strategy that can reduce worries about pain by informing the individual mechanism of the experienced pain and is considered essential before starting physical activity interventions [81]. However, although it has been proven effective in several chronic pain populations, the evidence in women with chronic pain in the pelvis is still nonexistent [101]. 

### Prevention of Chronic Pain in the Pelvis

During the woman’s lifespan there are windows of opportunity to prevent the transition from localized and periodic pain in the pelvis (dysmenorrhea or from pregnancy) into chronic pain (Figure 1). 

Pre-existing pro-inflammatory states increase the risk of chronification of pain [102]. During adolescence [103], and in relation to pregnancy [104], comorbid disorders of the urogenital systems are common and require attention. Early pain management with behavioral changes may reduce the risk of the transition towards chronic nociplastic pain in such cases, as was suggested from a systematic review (based on low to very low-quality evidence) [94]. 

Regarding the risk of chronic PGP, women at risk can be identified during pregnancy by a clinical assessment. The number of positive pain provocations tests is an established predictor of chronic PGP [8,105]. Pain provocation tests, as measures of increased pain sensitivity, have also been suggested as indicators of systematic inflammation [90]. Widespread pain, a characteristic feature of central sensitization [106], has been identified as the strongest predictor for a poor long-term outcome in women with PGP [107]. Moreover, it is reported that eleven years after PGP onset, women with chronic PGP show more concern and depression then women who recover from the PGP after pregnancy [8]. This observation supports the idea of cognitive-emotional sensitization in women with chronic PGP [108]. Therefore, clinical assessment in women with pregnancy-related PGP needs to contain at least pain provocation tests: a pain drawing to assess widespread pain and an evaluation of concern and depression [109,110]. This way, women at risk for chronic pain are identified and preventive interventions targeting the appropriate underlying mechanism might avoid the development of long-term pain [111].

The promotion of a healthy physical activity level during pregnancy, when women are extra prone to lifestyle changes [112], might improve health-promoting physical activity after childbirth [113]. Importantly, mothers’ habits, including a healthy lifestyle, have demonstrated a positive effect on the offspring, as healthy role modelling behaviors for infants [74,87,112]. Current guidelines advise healthy pregnant women to follow the general guidelines of weekly, evenly distributed, physical activity which last 150 min, performed at a moderate intensity, plus two times per week resistance training [87]. Due to anatomical and physiological changes, as well as foetal requirements, some modification of exercise habits may be necessary. Reported barriers related to physical activity after pregnancy have been related to capability (e.g., limitation in healthcare providers’ skills in providing lifestyle support), opportunity (support from partners) and motivation (e.g., identifying benefits of exercise) [74]. Therefore, it is important to focus on these barriers during pregnancy, and to find solutions on how to remove those obstacles [80]. In this context, it is known that women with PGP having a lack of knowledge and lack of support and knowledge from healthcare providers when seeking care, experience unmet needs [10,114,115]. Indeed, education and advice have been reported to positively influence pain, disability and/or sick leave [116]. 

It is common for pregnant women to seek knowledge and advice on the internet. However, it has been shown that bad advice flourishes and increases worries [117]. Therefore, women need guidance in what knowledge to trust and how to individualize it for their specific situation. This underlines the necessity for physical activity interventions and exercise regimes in pregnant women, guided by health care professionals with adequate education [86]. Indeed, expert advice and experiences on therapeutic exercise (with or without co-interventions) during pregnancy is proven effective [118]. In clinical practice, transcutaneous electric nerve stimulation and belts are common tools to support pain self-management, with some evidence to support their use in PGP [4,9]. They are generally seen as pain reduction tools but would rather be seen as tools to encourage appropriate physical activity, as they enable physical activity at a tolerable pain level. 

Lastly and importantly, the promotion of a healthy lifestyle does not exclude the assessment of local nociception-inducing mechanisms in clinical examination and medical assessment, in order to identify and treat specific symptoms related to disorders causing pain in the pelvis. 

## 6. Future Directions for Research

Women with pelvic pain due to severe dysmenorrhea are currently often told that their pain should be considered normal. In women with PGP during pregnancy or after delivery, pain is often considered a normal consequence of pregnancy and childbirth. Due to this narrow view on pain management, many women are undertreated for complaints that can be treatable. Since pain in the pelvis is associated with suffering, disability and a low health-related quality of life [115,119,120], there is an urgent need to better understand how to alleviate the suffering of women with pain in the pelvis, and to gain knowledge on adequate strategies to prevent (chronic) pain in the pelvis. This state-of-the-art review clearly shows that firm evidence on the role of lifestyle on pain, and on the effectiveness of lifestyle interventions on pain symptoms in women with pain in the pelvis, are essentially lacking. Therefore, we present a research agenda (Figure 2). 

### 6.1. Pain Neuroscience Education: A Prerequisite for Sustained Lifestyle Adaptations in Women with Pain in the Pelvis?

It could be theorized that teaching women the science behind pain and the mechanisms related to pain in the pelvis (such as lifestyle factors) could be a strong protective factor for developing chronic pain in the pelvis throughout life and for intrinsically motivating them for a sustained engagement in a healthy lifestyle. Thereby, pain neuroscience education in the treatment of women with pelvic pain might provide women in pain with the necessary information to reach a sustained change towards a more active and healthier lifestyle. However, what pain neuroscience education in women with pelvic pain should look like, and whether it is effective in changing beliefs and behaviors, and subsequently in decreasing pain, should be a primary topic of research. 

In this context, many questions related to pain education in women need to be considered. When during lifespan would it be optimal to educate women on pain which can appear in relation to menstruation or in relation to pregnancy and at postpartum? This question is of relevance as menstruation and pregnancy should not be medicalized, nor should be neglected. The delivery mode of pain education related to menstruation and pregnancy should also be considered. Can it be added to the curriculum in secondary school, or should the message be spread at a societal level to reach all stakeholders? In this context, it is also pertinent to study the societal and health care providers’ ideas and beliefs about pain in the pelvis. A reflection on which women would need to be educated is another relevant topic in this regard; all women versus women experiencing painful menstruation or PGP, versus women who are mothers experience pelvic pain? Furthermore, the content of the education to achieve the highest impact on women should be explored. Information on all former topics can be studied using quantitative and qualitative research approaches. This way, the barriers for optimal pain care on a patient’s, healthcare provider’s and societal level can be identified, and necessary content for pain neuroscience education for women with pain in the pelvis can be defined. 

### 6.2. A Broad View on Pain in Women with Pain in the Pelvis

Young menstruating women’s pain experiences need to be better understood in the development of disability and chronic pain in the pelvis. A better understanding of the role of multiple factors, such as lifestyle, genetic, psychosocial and patho-anatomical factors as predictors for severe pain in the pelvis would create opportunities for developing early preventive and curative strategies. The underlying effects related to lifestyle interventions in endometriosis-related pelvic pain, such as an estrogen-dependent and an anti-inflammatory effect, need to be further explored. Such information is necessary to develop the specific content for pain neuroscience education programs in women with pain in the pelvis. 

In the context of pregnancy-related pain, the pregnancy period is a period characterized by sleep deprivation, hormonal and psychological emotional changes, and less physical activity. For women after childbirth, pelvic floor trauma is an additional challenge in relation to lifestyle, besides those factors mentioned during pregnancy that all may prevail after pregnancy. Through shared neurophysiological mechanisms, these factors may contribute to the development of PGP in pregnancy and chronic PGP which lasts at postpartum. In this, the dominant pain mechanism in pregnancy-related PGP needs to be better understood. Since widespread pain, a characteristic feature of central sensitization [106], has been identified as the strongest predictor of poor long-term outcome in women with PGP [107], more studies on the role of central sensitization in the development of chronic PGP are needed to develop preventive strategies and individually tailored interventions (e.g., personalized pain education and behavior treatment). 

Furthermore, the potential of lifestyle adaptations, adapted to pregnancy, as an opportunity for preventing and decreasing the impact of PGP after pregnancy must be explored. Research on how current treatments for lifestyle interventions such as cognitive behavioral therapy as a sleep intervention [121], can be adapted specifically for women with pain in the early time periods following childbirth, is of high priority. Indeed, women ask for more knowledge and support on strategies to (self-)manage pain in relation to pregnancy [10,115]. Therefore, studies on the content and delivery method of pain management programs in women with pregnancy-related pain in the pelvis are needed, i.e., how can treatment be implemented with potential barriers to approaching a healthy lifestyle (e.g., common sleep deprivation due to pregnancy or young children)? In terms of delivery form, management by means of mobile-and-internet-delivered programs for prevention, as well as pain management programs, must be further explored. 

It is of interest to explore which barriers towards a healthy lifestyle exist in women with pain in the pelvis. It is, for example, of interest to explore whether transcutaneous electric nerve stimulation can be used as a nonpharmacological treatment to influence central sensitization in women with cyclic or periodic pain [122], and be an effective treatment for staying active and exercising during painful periods. Pelvic floor disorders as a consequence of childbirth need to be identified and managed, not only for the problems themselves but also as barriers to exercise. It is known that pregnancy and vaginal delivery are among the main risk factors for urinary and faecal incontinence [104], and that urinary incontinence affects exercise participation in one in two symptomatic women. Pelvic floor disorders including urinary and faecal incontinence, and pelvic floor prolapse are reported barriers to exercise with a moderate or great effect in 39% of women (95% Cl: 22%, 57%) [123]. Therefore, it is clear that women with pelvic pain should always be approached in a multidimensional way, by a multidisciplinary team. Indeed, chronic pain in the pelvis requires competences from multiple disciplines that involve physiotherapists, midwives, gynecologists, psychologists, occupational therapists, and pain specialized medical doctors.

Finally future research on PGP should consider the recent published core outcome set of PGP to enable future compilation of results in systematic reviews [110]. To strengthen the quality of the evidence, future studies would benefit from designing and presenting results in accordance with accepted reporting standards (e.g., CONSORT and PRISMA).

## 7. Conclusions

During their lifespans, many women are exposed to pain in the pelvis in relation to menstruation and pregnancy, which is often considered normal and inherently linked to being a woman. This leads to insufficient treatment. Severe dysmenorrhea, as seen in endometriosis and pregnancy-related PGP, has a great impact on daily activities, school attendance and work ability. Lifestyle factors such as a low physical activity level, poor sleep, or periods of (di)stress are all common in these challenging periods of women’s life.

Based on this state-of-the-art review, encouraging women with pain in the pelvis to be physically active and to exercise is supported to some extent. Clinicians are suggested to use a window of opportunity to prevent a potential transition from localized or periodic pain in the pelvis (e.g., pain during pregnancy and after delivery or severe dysmenorrhea) towards persistent chronic pain, by encouraging a healthy physical activity level and applying appropriate pain management. The available evidence on lifestyle factors such as sleep, (di)stress, diet, and tobacco/alcohol use is, however, inconclusive; very few studies are available, and the studies which are available are of general low quality. 

Research to disentangle the relationship between lifestyle factors, such as physical activity level, sleep, diet, smoking, and psychological distress, and the experience of pain in the pelvis is highly needed. Studies which address the development of management strategies for adapting lifestyle, which are specifically tailored to women with pain in the pelvis, and take hormonal status, life events and context into account, are required. 

## Figures and Tables

**Figure 1 jcm-10-05397-f001:**
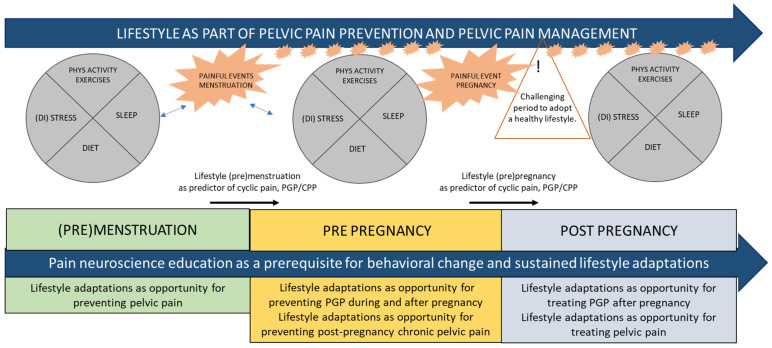
Women are exposed to pain in the pelvis during their lifespan in relation to hormonal changes and pregnancy. Lifestyle can influence the pain experience during the lifespan.

**Figure 2 jcm-10-05397-f002:**
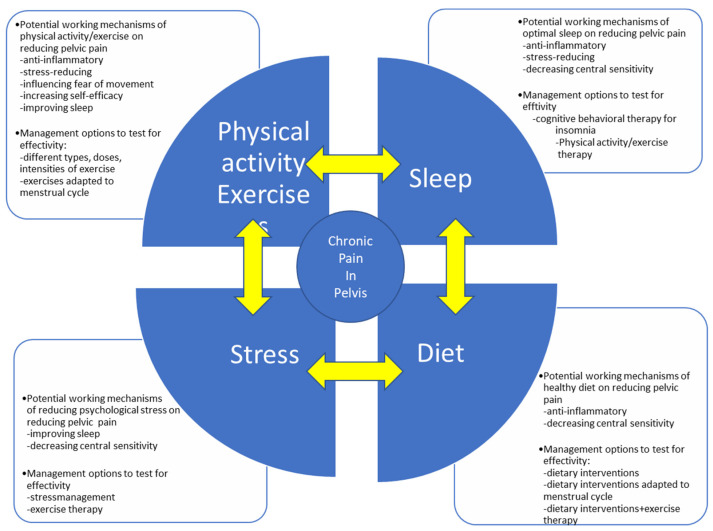
Research agenda of the potential working mechanisms of lifestyle factors on pain reduction in chronic pain in the pelvis, in parallel with potential management options.

**Table 1 jcm-10-05397-t001:** Best evidence for lifestyle factors for women with endometriosis according to systematic reviews following PRISMA.

Lifestyle Factor	First Author Year Published Type of Data Synthesis	Included Population	Number of Papers Included and Number of Participants in Original Research	Detail of Lifestyle Factor/Intervention Assessed	Quality of Studies (Tool and Results)	Main Results in Context of the Specified State-of-the-Art Objectives	Summary of Evidence
Physical activity and/or Exercise	Hansen et al., 2021 [43] Systematic review, narrative synthesis of data	Women of reproductive age with a laparoscopically confirmed diagnosisof endometriosis	Two RCTs (*n* = 39 and 40) Two cohort studies (*n* = 20 and 26) One case-control study (*n* = 81) One cross-sectional survey study (*n* = 484)	Any type of exercise	Cochrane risk of bias Tool for RCTs: ‘Unclear’ in at least 4 of 7 domains. Low risk of selection and attrition bias ROBINS-I quality assessment scale for case-control and cohort studies: moderate to critical risk of bias in at least 4 of 7 domains in 3 studies. 1 study with low risk of bias on 4 domains and no information on 2 domains	RCTs: no significant effect on pain Cohort studies: no consistent outcome: one positive impact on pain, one no impact on pain Case control study and survey: one no impact and one negative impact on pain	No indication for beneficial effect of exercise on pain in women with endometriosis
Physical activity and/or Exercise	Ricci et al., 2016 [44] Systematic review and meta-analysis	Women with a clinical and/or histological based diagnosis of endometriosis. However, all but one study included participants with laparoscopic confirmed endometriosis	Three cohort studies (*n* between 1481 and 2703) Six case–control studies (*n* cases between 50 and 268)	Recent and past recreational PA, evaluated in different ways: hours of PA per week, metabolic equivalents (MET)-h/week, author-defined low or high intensity activity	Newcastle-Ottawa quality assessment scale, 3 domains (selection –max. score 4; comparability—max score 2; exposure—max. score 3) case–control studies: between 2–4 on selection; 0–2 on comparability; 1–2 on exposure cohort studies: between 2–3 on selection; 2 on comparability; 1–2 on outcome	When adjusted estimates of odds ratios were provided, PA was protective against endometriosis. However, the overall estimate was not statistically significant when including all retrieved articles. Studies on the association between PA during adolescence and endometriosis are inconsistent, and meta-analysis’ results are inconclusive	Inconclusive results on relation between risk of endometriosis and PA
Diet	Huijs and Nap 2020 [45] Systematic review, narrative synthesis of data	Women with surgically or magnetic resonance imaging/ultrasound confirmed endometriosis	Four RCTs (*n* between 39–240) Four non-randomized clinical trials (*n* between 4–60) One retrospective study (=59) Once case series (*n* = 8) Two case reports (*n* = 2 and 1)	Nutrient or diet	GRADE criteria: low to very low Risk of bias Imprecision, inconsistency, indirectness and/or publication bias were found in all included studies	In nine studies, nutrients were added to patients’ diets, and in seven of these a positive effect was found. In three studies, nutrients were avoided, with positive effects on endometriosis associated symptoms	Dietary interventions may potentially have an influence on symptoms in women with endometriosis, but no clinical recommendations can be provided yet.
Diet	Nirgianakis et al., 2021 [46] Systematic review, narrative synthesis of data	Women diagnosed with endometriosis—no details described	Nine human studies, Two RCTs (*n* = 19 and 37) Two controlled studies (*n* = 30 and 35) Four uncontrolled before-after studies (*n* between 47 and 295) once qualitative study (*n* = 12)	Dietary intervention, including supplementation with selected dietary components, exclusion of selected dietary components, and complete diet modification.	Quality in Prognostic Studies tool for observational research Cochrane risk-of-bias tool for RCTs Moderate to high risk of bias	All included studies assessed a different dietary intervention, with most of them finding a positive effect on endometriosis.	No subcategories of patients who are more likely to benefit from a dietary intervention can be defined. No specific dietary interventions that ameliorate certain endometriosis-associated symptoms can be identified.
Diet	Qi et al., 2021 [47] Systematic review and meta-analysis	Women with a clinical and/or laparoscopic confirmed endometriosis. All but two case-control studies included women with laparoscopic confirmed endometriosis.	Two cohort studies (*n* = 581 and 1385) Five case-control studies (*n* between 78 and 504)	Dairy intake	Newcastle-Ottawa Scale, 3 domains (selection –max. score 4; comparability—max score 2; exposure—max. score 3) All studies scored 6 stars or higher and were considered high-quality studies.	Total dairy intake is inversely associated with the risk of endometriosis Risk of endometriosis tended to decrease when dairy products intake was over 21 servings/week (RR 0.87, 95% CI 0.76–1.00; *p* = 0.04). When more than 18 servings of high-fat dairy products per week are consumed, a reduced risk of endometriosis (RR 0.86, 95% CI 0.76–0.96) is reported. Stratified-analyses based on specific dairy product categories: Evidence for reduced risk of endometriosis: a high cheese intake (RR 0.86, 95%CI 0.74–1.00). No Evidence for reduced risk of endometriosis: High intake of whole milk (RR 0.90, 95% CI 0.72–1.12), reduced-fat/skim milk (RR 0.83, 95% CI 0.50–1.73), ice cream (RR 0.83, 95% CI 0.50–1.73), and yogurt (RR 0.83, 95% CI 0.62–1.11) Evidence for higher risk of endometriosis: Higher butter intake (1.27, 95% CI 1.03–1.55).	Dairy products intake is associated with a reduction in endometriosis, with significant effects when the average daily intake is three servings or more. When analyzed according to the specific type of dairy product, it is suggested that women with higher high-fat dairy and cheese intake have a reduced risk of endometriosis
(Di)-stress	Evans et al., 2019 [48] Systematic review, narrative synthesis of data	Women with medically confirmed endometriosis (such as via a previous laparoscopy)- no further details	Three RCTs (n between 28 and 50) One qualitative study (*n* = 28) One controlled study (*n* = 64) Two single arm studies (*n* = 26 and 10) One retrospective cohort study (*n* = 47) Two case series (*n* = 5 and 2)	Physical therapy with cognitive behavioral therapy, yoga, biofeedback, mindfulness and psychotherapy, psychotherapy combined with acupuncture, and progressive muscle relaxation	Four criteria of the Cochrane Risk of Bias tool for RCTs (adequate generation of allocation sequence, concealment of allocation to conditions, assessor masking, dealing with incomplete data): For RCTs, mostly low risk of bias, apart from attrition bias and selection bias For non-randomized trials (including single-arm studies), observational cohort studies and case reports, the relevant National Institutes of Health quality assessment tool was used with a final quality rating of ‘Good,’ ‘Fair’ or ‘Poor.’: Fair risk of bias in all non-randomized trials Critical Appraisal Skills Programme for qualitative studies (10 questions which can be answered “Yes”, “Can’t Tell”, or “No.”, with “Yes” implying a low risk of bias): Low risk of bias	89% of studies report improvement in pain	Based on the included mainly pilot studies, it is suggested that psychological and mind-body interventions show promise in alleviating pain in women with endometriosis
Tobacco use	Bravi et al., 2014 [49] Systematic review and meta-analysis	Women with histologically confirmed and/or clinically based diagnosis of endometriosis.	Nine cohort studies (*n* between 19 and 3110) Twenty-nine case-control studies (*n* between 28 and 947)	Tobacco smoking	Newcastle-Ottawa Scale, 3 domains (selection –max. score 4; comparability—max score 2; exposure –max. score 3) Cohort studies: 2 studies 6, rest lower Case control: 3 more than 6, other below 6 Most studies have high risk of bias	Considering ever smokers or, separately, former smokers, current smokers, moderate smokers and heavy smokers, no statistically significant association is found with risk of endometriosis	There is no association between smoking and risk of endometriosis

RR: risk ratio; SMD: standardized mean difference; CI: 95% confidence interval; SR: systematic reviews; RCT: randomized controlled trials; PA: physical activity.

**Table 2 jcm-10-05397-t002:** Best evidence for lifestyle interventions for women with pelvic girdle pain according to systematic reviews following PRISMA.

Lifestyle Factor	First Author Year Published Type of Data Synthesis	Included Population PGP/LPP/LBP	Pregnancy/ Postpartum Number of SR/RCT Number of Participants	Detail of Lifestyle Factor/Intervention Assessed	Quality of Studies (Tool and Results)	Main Results on Interventions	Summary of Evidence
Physical activity and/or Exercise	Weis et al., 2020 [55] SR and meta-analysis	PGP/LPP LBP	Pregnancy Six SRs based on nine RCTs **PGP**: Four SRs based on three RCTs **LPP**: Nine SRs based on four RCTs **LBP**: Five SRs based on seven RCTs	land or water-based exercise in group or individual	Modified version of Scottish Intercollegiate Guideline Network **PGP**: high-quality, low risk of bias (1 SR) acceptable-quality, moderate risk of bias (1 SR) low-quality, high risk of bias (2 SR) **LPP**: high-quality, low risk of bias (1 SR) acceptable-quality, moderate risk of bias (7 SR) low-quality, high risk of bias (1 SR) **LBP**: high-quality, low risk of bias (1 SR) acceptable-quality, moderate risk of bias (3 SR) low-quality, high risk of bias (1 SR)	**PGP**: decreased pain intensity and disability **LPP**: improved function and reduced prevalence from most SRs **LBP**: 4/5 SRs reduced pain intensity and disability	**PGP**: inconclusive, evidence with favorable outcomes **LPP**: moderate strength evidence with unclear outcomes **LBP**: inconclusive, strength with favorable outcomes
Physical activity and/or Exercise	Weis et al., 2020 [42] SR and meta-analysis	PGP/LPP	Postpartum Two SRs based on six RCTs **PGP**: Two SRs based on four RCTs **LPP**: Two SRs based on two RCTs **LBP**: 0 SR	exercise in group or individual	Modified version of Scottish Intercollegiate Guideline Network **PGP**: Acceptable-quality, moderate risk of bias **LPP**: Acceptable-quality, moderate risk of bias	**PGP**: One of three SRs stated additional effect of exercises at reducing pain and disability. **LPP**: no firm conclusions could be drawn	**PGP**: moderate strength of evidence with unclear outcomes. **LPP**: inconclusive strength of evidence and unclear outcomes
Physical activity and/or Exercise	Davenport et al., 2019 [56] SR and meta-analysis	PGP/LPP LBP	Pregnancy and postpartum Thirty-two studies (*n* = 52297) out of 23 RCTs (13 exercise only and 10 exercise +co-interventions) **pregnancy** -pooled estimates prevalence 12 RCTs (*n* = 1987) -pooled estimates severity 10 RCTs (*n* = 784). **postpartum** -pooled estimates prevalence 3 RCTs (*n* = 491) -severity 1 RCTs (*n* = 257)	result presented from ‘exercise only’ or ‘exercise with co-intervention’ (yoga, aerobic exercise, general muscle strengthening or muscle strengthening specific to one body region and combination of aerobic and resistance training)	GRADE criteria **pregnancy** Prevalence: very low-quality Serious risk of bias, serious directness of the interventions and serious imprecision. Severity: very low-quality Serious risk of bias, serious inconsistency, and serious indirectness of the interventions. **postpartum**: Prevalence: low-quality Serious indirectness of the interventions and serious imprecision. Severity: low quality Serious risk of bias and serious inconsistency.	**pregnancy** PGP/LPP/LBP: no reduced odds (OR 0.78, 95% CI 0.60, 1.02) lower pain severity (standardized mean difference −1.03, 95% CI −1.58, –0.48) **postpartum** PGP/LPP/LBP: no reduced odds (OR 0.89, 95% CI 0.51, 1.56) decreased severity of LBP (*p* = 0.034) (only 1 RCT)	**pregnancy** PGP/LPP/LBP: very low to moderate quality evidence **postpartum** PGP/LPP/LBP: low quality to moderate evidence low quality evidence
Physical activity and/or Exercise	Almousa et al., 2018 [57] SR	PGP	Pregnancy and postpartum Five RCTs + 1 follow up (n between 44 and 330) **pregnancy** Two RCTs (*n* = 426) **postpartum** Two RCTs (*n* = 125) +1 follow up (*n* = 65) pregnancy and postpartum One RCT (*n* = 103)	stabilizing exercises	Pedro scale Scores range 5–8, i.e., 5 studies good-quality and 1 study fair-quality	**pregnancy** decrease pain (2 RCTs) and improved quality of life (1 RCT) **postpartum** contradictory results	insufficient evidence
Physical activity and/or Exercise	Shiri et al., 2018 [58] SR and meta-analysis	PGP LBP	Pregnancy pooled 11 RCTs (*n* = 2347) **PGP**: Four RCTs (*n* = 565) **LPP**: Eight RCTs (*n* = 1737); out of only three RCTs (*n* = 1168) evaluating sick leave) **LBP**: Seven RCTs (*n* = 1175) out of only two RCTs (*n* = 349) evaluating sick leave	exercise of different type and in different combinations	Cochrane Collaboration’s tool. Low heterogeneity in meta-analyses * Low publication bias *	**PGP**: no protective effect (RR 0.99, 95% CI 0.81–1.21 **LPP**: no protective effect (RR 0.96, 95% CI 0.90–1.02) but prevented new episodes of sick leave (RR 0.79, 95% CI 0.64–0.99) **LBP**: reduced risk 9% (pooled RR) 0.91, 95% CI 0.83–0.99) prevented new episodes of sick leave (RR 0.67, 95% CI 0.40–1.12)	Exercise appears to reduce risk of LBP and sick leave due to LPP but no clear evidence on PGP
Physical activity and/or Exercise	Liddle and Pennick 2015 [4] SR and meta-analysis	PGP/LPP LBP	pregnancy Thirty-four RCTs (*n* = 5121) **PGP**: Six RCTs (*n* = 889) pooled two RCTs (*n* = 374) **LPP**: Thirteen RCTs (*n* = 2385) pooled four RCTs (*n* = 1176) **LBP**: Fifteen RCTs (*n* = 1847) pooled seven RCTs (*n* = 645)	exercises on land/in water	Cochrane Collaboration’s tool GRADE criteria Clinical heterogeneity precluded pooling results in many cases. Statistical heterogeneity was substantial in all but three meta-analyses, not improving following sensitivity analyses. Publication bias and selective reporting cannot be ruled out.	**PGP**: no effect on prevalence of PGP **LPP**: reduced prevalence and pain (RR 0.66; 95% CI 0.45 to 0.97) reduced sick leave (RR 0.76; 95% CI 0.62 to 0.94.) **LBP**: no effect on prevalence (RR 0.97; 95% CI 0.80 to 1.17) reduced pain (SMD −0.64; 95% CI −1.03 to −0.25) and reduced functional disability (SMD −0.56; 95% CI −0.89 to −0.23)	**PGP**: Low quality evidence **LPP**: Moderate-quality evidence Low-quality evidence **LBP**: Low-quality evidence
Physical activity and/or Exercise	Tseng et al., 2015 [59] SR	LPP	postpartum Four RCTs *n* = 251	exercise programs to strengthen deep local muscles and global muscles in the lumbopelvic regions	Cochrane Collaboration’s tool Pedro scale Scores range 4–8, i.e., three studies good-quality and one study fair quality. All studies except one were at low risk of bias on key domains such as sequence generation, allocation concealment, blinding of participants and personnel, completeness of outcome data for each main outcome, and selective reporting.	inconclusive on pain intensity and disability	no evidence

PGP: pelvic girdle pain; LPP: lumbopelvic pain; LBP: low back pain; RR: risk ratio; SMD: standardized mean difference; CI: 95% confidence interval; SR: systematic reviews; RCT: randomized controlled trials; * for details on risk of bias, refer to original study.

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
