# Peer review of "Lifestyle and Chronic Pain in the Pelvis: State of the Art and Future Directions"

_jcm, 2021, doi:10.3390/jcm10225397_

Round 1

Reviewer 1 Report

Very good systemic review 

But majority of findings are negative or inconclusive, which makes us confused.

1:  Endometriosis isn’t always the culprit of Chronic Pain in the Pelvis.
Chronic pelvic inflammatory disease is another major cause of  Chronic Pain in the Pelvis. The authors did not discuss this issue.

2: Relieving Pelvic Pain During and after Pregnancy: Pregnancy weakens the ligaments that keep the pelvic bones together. If those weakened ligaments become overloaded or injured, it results in pelvic instability. It is very famous idea but the authors did not mention about it.

Author Response

We thank the reviewers for valuable comments that have improved our manuscript ID: jcm-1438603 and that gave us the possibility to clarify some points.

A point-to-point list of replies to the reviewers’ comments

Reviewer 1:

1:  Endometriosis isn’t always the culprit of Chronic Pain in the Pelvis.
Chronic pelvic inflammatory disease is another major cause of  Chronic Pain in the Pelvis. The authors did not discuss this issue.

Reply: Thank you for this valuable and important comment. There are many possible causes for pain in the pelvis. In this review, we focused on two common disorders related to specific painful event during lifespan in women to point out the possibility to prevent development of chronic pain in women by addressing lifestyle factors. We have tried to clarify this on page 2, lines 82-84.

2: Relieving Pelvic Pain During and after Pregnancy: Pregnancy weakens the ligaments that keep the pelvic bones together. If those weakened ligaments become overloaded or injured, it results in pelvic instability. It is very famous idea but the authors did not mention about it.

Reply: Thank you for another valuable comment, that gave us the possibility to clarify this point. That pregnancy-related hormones have an impact on the ligaments of the pelvis is very true. However, since there is no firm consensus on whether this results in a pelvic instability that causes pelvic girdle pain in pregnancy, we have added and clarified it as a possible cause. Page 3, line 143.

Reviewer 2 Report

Well done on producing this 'state of the art' paper on lifestyle interventions for pelvic pain. My comments relate to bigger picture aspects of the paper and how this paper fits into the current literature.

Firstly, it would be good for you to point out in the introduction, why this is not a systematic review. You have gone to so much effort to produce this large peice of work, yet is is only a 'best evidence' review. This downgrades it's value in the evidence hierachy.

It would also add to the paper if you could comment on where research into lifestyle factors fits in with other research into pelvic pain. And in your Discussion, make comment on reasons why the evidence for each lifestyle factor is so poor - these two issues are interrelated. This may be out of the scope of your review, but comments on these in the introduction or discussion would add to the context of this paper.

Your final sentence (line 737-740) in the conclusion goes beyond what you have covered in your paper. This should be deleted.

This is a long paper. It may be too long to be published in its current form. Some editing of it would be useful to reduce its length. There are also English edits that would improve the paper.

Author Response

We thank the reviewers for valuable comments that have improved our manuscript ID: jcm-1438603 and that gave us the possibility to clarify some points.

A point-to-point list of replies to the reviewers’ comments

Reviewer 2:

Well done on producing this 'state of the art' paper on lifestyle interventions for pelvic pain. My comments relate to bigger picture aspects of the paper and how this paper fits into the current literature.

Firstly, it would be good for you to point out in the introduction, why this is not a systematic review. You have gone to so much effort to produce this large peice of work, yet is is only a 'best evidence' review. This downgrades it's value in the evidence hierachy.

Reply: Since there were so few studies within lifestyle factors in relation to chronic pelvic pain, a systematic review and meta-analysis would give no strong evidence to present. Furthermore, we wanted to link the fields of two commons pain disorders during women’s lifespan, that are not commonly looked into together, since they would potentially be a source of chronic pain. We believe ‘the best evidence format’ was most suitable to present an elaborated research agenda for this greater overall picture of chronic pain development in women. The focus is on pain development into chronic phase, not different causes of pelvic pain. We have clarified on page 4, lines 190-193.

It would also add to the paper if you could comment on where research into lifestyle factors fits in with other research into pelvic pain. And in your Discussion, make comment on reasons why the evidence for each lifestyle factor is so poor - these two issues are interrelated. This may be out of the scope of your review, but comments on these in the introduction or discussion would add to the context of this paper.

Reply: Up to recently, the research paradigm has had a large focus on the ‘bio’ part in the bio-psycho-social model. However, in this paper we have put pelvic pain within a broader perspective and call for a bio+psycho+social perspectives. Please see page 4 under the section 1.3 Lifestyle factors in Chronic Pain in the Pelvis.
From chronic pain science, it is known that physical activity might have a sleep improving, stress-reducing and general anti-inflammatory effect, which are all relevant for optimal pain management (please see page 19). Researchers within the field of pain in the pelvis thus need to explore the lifestyle factors’ relation to pain in the pelvis and potential preventive and curative strategies which our paper hope to stimulate.

Regarding why the evidence situation is poor on the scope of this paper, we can only speculate. It is only recently, that modern pain management included the management of lifestyle factors. Likewise, the calls for actions such as the World Health Organization’s (WHO’s) recommended stronger focus towards a healthy lifestyle, i.e. being physical active for a better health and for preventing non-communicable diseases as well as the recent action plan for the prevention of long-term disability from chronic pain in general across the lifespan, ‘a positive health concept’ as overarching strategic approach, is also in recent years (please see page 18). Hopefully, this paper will stimulate to increased research and thereby evidence within the field.

Your final sentence (line 737-740) in the conclusion goes beyond what you have covered in your paper. This should be deleted.

Reply: we believe that these sentences conclude what our paper is about for clinicians i.e. that there is no strong evidence for lifestyle interventions in endometriosis and pelvic girdle pain BUT there is strong call for clinicians to be part of the overall improvement of general health by addressing lifestyle factors grounded on strong evidence. We have revised the Conclusion and now hope it is better interpreted.  

This is a long paper. It may be too long to be published in its current form. Some editing of it would be useful to reduce its length. There are also English edits that would improve the paper.

Reply: we have done the minor language editing and a second spell check. We now hope this is to your satisfaction.
